# *Xenopus laevis* Oocyte Array Fluidic Device Integrated with Microelectrodes for A Compact Two-Electrode Voltage Clamping System

**DOI:** 10.3390/s23052370

**Published:** 2023-02-21

**Authors:** Nobuo Misawa, Mitsuyoshi Tomida, Yuji Murakami, Hidefumi Mitsuno, Ryohei Kanzaki

**Affiliations:** 1School of Veterinary Medicine, Azabu University, 1-17-71 Fuchinobe, Chuo-ku, Sagamihara 252-5201, Kanagawa, Japan; 2Department of Electrical and Electronic Information Engineering, Toyohashi University of Technology, 1-1 Hibarigaoka, Tempaku-cho, Toyohashi 441-8580, Aichi, Japan; 3Department of Electrical and Electronic Engineering, Faculty of Science and Technology, Shizuoka Institute of Science and Technology, 2200-2 Toyosawa, Fukuroi 437-8555, Shizuoka, Japan; 4Research Centre for Advanced Science and Technology, The University of Tokyo, 4-6-1 Komaba, Meguro-ku 153-8904, Tokyo, Japan

**Keywords:** *Xenopus laevis* oocytes, olfactory receptor, ion channel current, two-electrode voltage clamping, fluidic device

## Abstract

We report on a compact two-electrode voltage clamping system composed of microfabricated electrodes and a fluidic device for *Xenopus laevis* oocytes. The device was fabricated by assembling Si-based electrode chips and acrylic frames to form fluidic channels. After the installation of *Xenopus* oocytes into the fluidic channels, the device can be separated in order to measure changes in oocyte plasma membrane potential in each channel using an external amplifier. Using fluid simulations and experiments, we investigated the success rates of *Xenopus* oocyte arrays and electrode insertion with respect to the flow rate. We successfully located each oocyte in the array and detected oocyte responses to chemical stimuli using our device.

## 1. Introduction

One advantage of chemical detection using biomaterials is their specificity to target substances. On one hand, biomaterials are unsuitable for long-term use, and maintenance of their native function is difficult owing to denaturation and degradation. This instability limits their applications as elements of durable chemical sensors. However, the use of biomaterials has attracted a great deal of attention based on their highly specific molecular recognition ability, which can be accomplished in extremely microscopic systems. Such substrate specificities of biomolecules are adequate for short-term use. Therefore, many researchers and engineers have developed and examined various chemical sensors using biomaterials [1,2,3,4,5,6]. For instance, DNA is a robust molecule in a variety of biomaterials, and oligonucleotides are useful as probe molecules for some screening and sensing technologies, such as DNA chips for gene expression profiling and DNA aptamers for chemical detection [7,8].

Many proteins function in enzymatic catalysis, and antigen–antibody reactions are also applied to biosensing technologies. Although water-soluble proteins are mainly used for sensing applications based on their specific ligand-binding properties, membrane proteins have gained recent attention owing to their further signal transduction benefits.

As bio-based chemical sensors, some membrane proteins are reconstructed in artificial cell membranes integrated with electric devices [9,10,11]. However, there is still a missing piece of the puzzle regarding the mechanisms of many receptors and ion channels. Partly for that reason, membrane proteins are often inactive, even when they are reconstructed in artificial cell membranes. Accordingly, there are still impediments in the use of membrane proteins in fully artificial membrane systems. However, many membrane proteins that are posteriorly expressed in some host cells by gene engineering function properly [12,13,14,15], and *Xenopus laevis* oocytes are stably used as host cells for insect olfactory receptor expression [16]. A two-electrode voltage clamping (TEVC) method is generally used for electrophysiological measurements of current shifts with changes in *Xenopus* oocyte membrane potential [17]. The conventional TEVC setup requires a large footprint; therefore, we attempted to downsize the TEVC system in order to develop a compact chemical sensor device. We previously reported fluidic chemical sensor devices using *Xenopus* oocytes that expressed olfactory receptors [18,19,20,21]. *Xenopus* oocytes are easy to handle owing to their cell size of approximately 1 mm in diameter. In particular, their spherical shape enables them to be easily rolled into fluidic channels. However, our first reported device required each oocyte to be manually set into a predefined position within the device due to the configuration of the glass capillary electrodes [18,19]. To overcome this laborious operation, we developed a fluidic device for *Xenopus* oocyte arrays integrated with microfabricated electrodes [20,21]. The device was initially designed exclusively for fluidic channel sharing for oocyte installation and the introduction of sample solution. Accordingly, it could not be used to apply multiple chemical stimuli to each oocyte.

In this study, we fabricated Si-based electrodes for TEVC with a fluidic device made of acrylic resin that was capable of simple electrode positioning. After the electrodes are inserted into oocytes, the device can be partitioned into a TEVC device unit that can measure changes in oocyte membrane potential. We examined oocyte arrays and the appropriate insertion of electrodes with respect to flow rate. As a model of membrane proteins, we used the artificially expressed silkmoth (*Bombyx mori*) olfactory receptor BmOR3, which could be expressed in *Xenopus* oocytes by gene engineering [22] and was sensitive to a silkmoth pheromone bombykal. We measured changes in oocyte membrane potential caused by ligand irritation to validate the effectiveness of this compact TEVC system.

## 2. Materials and Methods

### 2.1. Device Design and Fabrication

The structure of the device is shown in Figure 1. The fluidic channel unit was made by combining two fabricated acrylic plates and one electrode chip. For the *Xenopus* oocyte array, trap and bypass channels were made following methods used for the previously reported dynamic bead array [23,24]. The top and bottom acrylic plates were adjusted and fixed to each other by surface contact and four magnets embedded in each plate. The width and height of all channels were 1.5 mm. To prevent the spilling of solution when units were dismounted from the concatenated fluidic device after oocyte installation, the oocyte trapping regions with electrodes were located 1.5 mm lower than the inlet and outlet channels, as shown in Figure 1b. The fluidic device was fabricated by machining a 5 mm thick poly (methyl methacrylate) (PMMA) substrate (Acrylic sheet, Sumitomo Chemical Co., Ltd., Tokyo, Japan) using an automated milling machine (Vertical Machining Centre S400, ENSHU Ltd., Hamamatsu, Japan) with a 0.5 mm diameter end mill (DLC-EDS, OSG Corp., Toyokawa, Japan).

The electrode chip was fabricated as follows. Both sides of the polished silicon substrate (Si(100), 350 ± 25 μm in thickness, Ryoko Sangyo Corp., Tokyo, Japan) were preliminarily cleaned with 2% hydrofluoric acid solution and the surface SiO_2_ layer of approximately 1 μm in thickness was prepared by wet oxidation. Then, the substrate was processed by the photoresist technique using double-sided mask aligner (PEM-800, Union Optical Co., Ltd., Tokyo, Japan), and double-sided etching was performed using a reactive ion etching (RIE) apparatus (L-451D-L, Canon ANELVA Corp., Kawasaki, Japan) and a deep RIE apparatus (MUC21-RD, Sumitomo Precision Products Co., Ltd., Amagasaki, Japan) to form cantilever-like structures. After removing the native SiO_2_ layer using 1% hydrofluoric acid solution for 10 s, the substrate was soaked in 25% tetramethylammonium hydroxide solution for 10 min at 80 °C to form sharply-angled electrode tips by anisotropic etching. The metal layers, Ti (50 nm thick) and Ag (800 nm thick), were continuously deposited on the SiO_2_ surface of the substrate by the multi-target sputter system (C-7250, Canon ANELVA Corp., Kawasaki, Japan). Poly(chloro-p-xylylene) (parylene-C) (approximately 700 nm thick) was deposited to insulate the entire substrate surface using a parylene coater (PDS 2010, SCS Inc., Indianapolis, IN, USA). Finally, the electrode tip areas and contact pads were selectively exposed by Ar plasma etching with the RIE system (CE-300I, ULVAC, Inc., Chigasaki, Japan). The Ag surface was altered to AgCl by dipping it in a sodium hypochlorite solution. The electrodes were located at the central portion of the flow path in trapping regions. The tip width of the electrodes was 10 μm, and the thickness was approximately 50 μm. Based on our previous study [19], the distance between the two electrode tips was 570 μm, and the angle between the electrodes was 30° to the left and right relative to the flow direction, as shown in the inset of Figure 1a. To prevent leakage of the solution from gaps between the PMMA parts and electrode chips, additional parylene-C (5 μm thick) was deposited on the entire assembled device except the contact pads and inner portions of the device.

### 2.2. Fluid Simulation

Using software for fluid analysis (Ansys 14.0 Fluid Flow CFX, Ansys Japan K.K., Tokyo, Japan), streams in the fluidic channel device were verified. The fluid was assumed to be water. The streams were simulated for a flow rate of 1–10 mL min^−1^ with and without *Xenopus* oocytes in the trapping region. In this simulation, *Xenopus* oocytes were expedientially assumed to be fixed spheres of 1.3 mm in diameter.

### 2.3. Xenopus Oocyte Array and Electrode Insertion

*Xenopus* oocytes were introduced into the fluidic channel by suction from the outlet using a syringe pump (HA1100W Pump 11 Elite, Harvard Apparatus, Holliston, MA, USA). The introduction of the oocytes into the fluidic device was performed by manually approaching each oocyte, one-by-one, with a micropipette at the inlet of the tube connected to the device’s inlet side. Several flow rates ranging 1–30 mL min^−1^ were tested. Barth’s solution (88 mM NaCl, 1 mM KCl, 0.3 mM Ca(NO_3_)_2_, 0.4 mM CaCl_2_, 0.8 mM MgSO_4_, 2.4 mM NaHCO_3_, and 15 mM HEPES, pH 7.6) was used for the flow. To investigate the oocyte array and electrode invasion level into oocytes, video images were recorded at 30 fps using a digital camera (EXILIM EX-F1, CASIO Computer Co., Ltd., Tokyo, Japan). The images were processed with Image-Pro Plus 6.0 (Media Cybernetics, Inc., Rockville, MD, USA). The appropriate insertion of electrodes was validated based on the micrographs and signal current recordings.

### 2.4. Receptor Expression and Electrophysiological Recording

Egg-bearing *Xenopus laevis* was raised in an incubator at 20 °C, and the oocytes were collected by laparotomy under ice-cold anesthesia. Stage V to VI oocytes with perceived diameters of 1.0–1.3 mm [25] were treated with 1.5 mg mL-1 collagenase in Ca^2+^-free saline solution (82.5 mM NaCl, 2 mM KCl, 1 mM MgCl_2_, and 5 mM HEPES, pH 7.5) for 1.5 h at 20 °C. Using a microinjector (Nanoject II, Drummond Scientific Company, Broomall, PA, USA), the oocytes were then injected with 25 ng of the olfactory receptor gene (BmOR3) RNA and the co-receptor protein gene (Or83b family gene) RNA synthesized using a mMESSAGE mMACHINE (Life Technologies, Carlsbad, CA, USA). The injected oocytes were incubated for 3 days at 20 °C in Barth’s solution without any antibiotics. Oocyte follicles were manually removed using tweezers before being installed in the fluidic device for TEVC.

The signal currents of the oocytes were recorded using the TEVC method with a custom-built amplifier (Triton, Tecella, LLC, Costa, Mesa, CA, USA). Extracellular fluid was grounded with an independent Ag/AgCl wire that was directly inserted into the fluidic channel. The currents were monitored every 100 μs at a hold voltage of −80 mV. The current traces were processed by an eight-pole Bessel low-pass filter with a cut-off frequency of 10 Hz.

As a model of chemical stimulus, 20 μL of solution was added manually, with and without 100 μM (*E*,*Z*)-10,12-hexadecadienal (bombykal) containing 1% dimethyl sulfoxide, into the fluidic channel using a pipette without perfusion.

## 3. Results and Discussion

Figure 2a shows the assembled device integrated with electrodes. The electrode substrate and fabricated PMMA parts could be aligned manually without microscopic observation. The TEVC device units (Figure 2b) were mounted to the bottom frame in contact with each other. We successfully observed oocyte motion in the fluidic channel owing to the transparency of PMMA. The solution flowed stably in the fluidic channel without leakage for a flow rate in the range of 1–30 mL min^−1^.

Based on the fluid simulations, we found that the bypass and trap channel flow rates changed depending on the entry flow rate in the fluidic channel. At a lower entry flow rate of 1 mL min^−1^, the main flow proceeded to the bypass channel and not to the trap channel (Figure 3a). The direction of flow to the trap channel was dominant for higher entry flow rates in the absence of oocytes at the trapping site (Figure 3a–c). Figure 3d shows subsequent oocytes flowed to the bypass channel in the presence of trapped oocytes that act as a plug for the trap channel.

As the side views of the simulations show, there were uneven flows in the depth direction that could not be estimated by simple two-dimensional streamline analyses. The simulation outcomes indicate that when the flow rate is sufficiently fast, an oocyte gets trapped in trapping sites that lack oocytes. Consequently, we considered that a proper flow rate enables all oocytes to be sequentially arrayed in order, upstream to downstream, in the fluidic device.

We experimentally investigated the success rate of oocyte trapping for various flow rates. As shown in the superimposed images taken at 0.5 s intervals in Figure 4a, an oocyte passed through the channel at a flow rate of 3 mL min^−1^ (Appendix A). In contrast, at a faster flow rate of 20 mL min^−1^, four different oocytes were uneventfully trapped in sequence (Figure 4b and Appendix A).

In oocyte trapping mode, subsequent oocytes occasionally touched the previously trapped oocytes when they turned toward the bypass channel at a fork in the fluidic channel. The observed oocyte motion was consistent with the results of the fluid simulation (Figure 3d). Figure 4c shows a graph of the relationship between flow rate and oocyte trapping success. A flow rate of greater than approximately 5 mL min^−1^ facilitated oocyte trapping. The success rate of oocyte trapping corresponded with the simulation results, and this relationship might be potentially affected by variations in the size of *Xenopus* oocytes used in the experiment [1.3 ± 0.1 mm (SD)]. At a flow rate of greater than 10 mL min^−1^, oocytes were consistently trapped. Thus, we employed a flow rate of greater than 10 mL min^−1^ for the oocyte array.

The success of trapping did not necessarily correspond to the successful insertion of electrodes into the trapped *Xenopus* oocyte. We examined the adequate flow rate to ensure the insertion of electrodes into the oocyte using flow rates exceeding 10 mL min^−1^. Figure 5 shows electrode invasion levels for oocytes that were categorized as contact, insertion, and penetration or pass. Figure 5b represents a successful insertion. The highest probability of insertion was approximately 70% and was observed at a flow rate of approximately 24 mL min^−1^.

Although the success rate of insertion did not reach 100% in this study, our previous study using glass capillary electrodes also resulted in an approximately 90% success rate using the same angle between the two electrodes [19]. Hence, the insertion success rates by flowing appear to plateau at around 70–90%. However, this corresponded to a flow rate that was much faster than that in our previous study [19] (approximately 2 mL min^−1^), which used glass capillary electrodes. We believe this difference can mainly be attributed to differences in the tip shape and size of electrodes in addition to the fluidic channel structure.

The tip of the glass capillary electrode for TEVC is generally a circle with an outer diameter of less than 1 μm. Our fabricated electrode tips had a polygonal shape with sides being tens of micrometers in length, even after sharpening by anisotropic etching. We determined that further sharpening while maintaining structural strength is required to increase the insertion rate to a level similar to that observed using glass capillaries. In fact, the success rate of insertions reached only approximately 30%, at a maximum, when we used unetched electrodes.

Figure 6 shows a representative current recording result for an oocyte expressing BmOR3 using our proposed TEVC system without perfusion. As a negative control, we first added and removed Barth’s solution without bombykal (Figure 6a,b). We observed no particular current changes except for spike noise (Figure 6b) caused by the pipette aspiration. Additionally, we successfully obtained a ligand-induced signal when we added bombykal solution (Figure 6c). We also detected spike noise (Figure 6d) due to pipette aspiration, in this positive control experiment.

The signal intensity, approximately 1 µA, was considerably lower than expected based on the high concentration of bombykal (20 µM as the final concentration in the fluidic device). This was presumably due to the expression level of BmOR3 in this experiment, and differences in the electrical properties between bare Ag/AgCl electrodes and commonly used glass capillary electrodes mediated by KCl solution.

In this experimental system, we did not use a perfusion device for the solution exchange, and for descriptive purposes, a highly concentrated bombykal solution was used to demonstrate the device properties. Excessive quantities of bombykal affected BmOR3 even after the pipette aspiration shown in Figure 6d, and the signal current did not return to the baseline level over short time periods, unlike in our previous work [19]. Consequently, we were not able to test the lifetime of this sensor device by repeated chemical stimuli. The system can be improved by further upgrading the electrodes in the future. Regarding the lifetime of the sensor using *Xenopus* oocytes, in the case of the glass capillary electrodes in our previous work [19], the activity was maintained for about half a day at room temperature. However, considering the current thickness of Si-based electrodes in this research, it is estimated that the lifetime of the sensor will be several hours at the longest; therefore, we think that it is suitable for more temporary sensing at present.

## 4. Conclusions

We constructed a *Xenopus laevis* oocyte array fluidic device made of acrylic resin integrated with Si-based microelectrodes for use as a compact TEVC system. Owing to the fabrication of electrodes by the photolithography process, assembly was far easier than that of previous devices. Additionally, in contrast to conventional cell array fluidic devices, our proposed device can be manually separated to discrete TEVC units after the installation of each oocyte. Oocytes were consistently arrayed at a flow rate of greater than 10 mL min^−1^ and the microelectrodes were successfully inserted into oocytes with about 70% probability at an entry flow rate of approximately 24 mL min^−1^. Moreover, using the fabricated device, we succeeded in detecting a silkmoth pheromone, bombykal, through an oocyte that expressed BmOR3 to demonstrate the feasibility of the device. The device was easy to operate and had a small footprint. Therefore, we expect that our proposed device will facilitate high-throughput chemical sensing and screening using *Xenopus* oocytes. Si-based electrodes are suitable for multi-channel measurement systems because they can be mass-produced simultaneously with uniform characteristics. In addition, since our fluidic devices have made it possible to apply multiple oocytes, we believe that in the future we will be able to detect multiple chemical substances with higher sensitivity by expanding the types of expression receptors and constructing highly expressing systems.

## Figures and Tables

**Figure 1 sensors-23-02370-f001:**
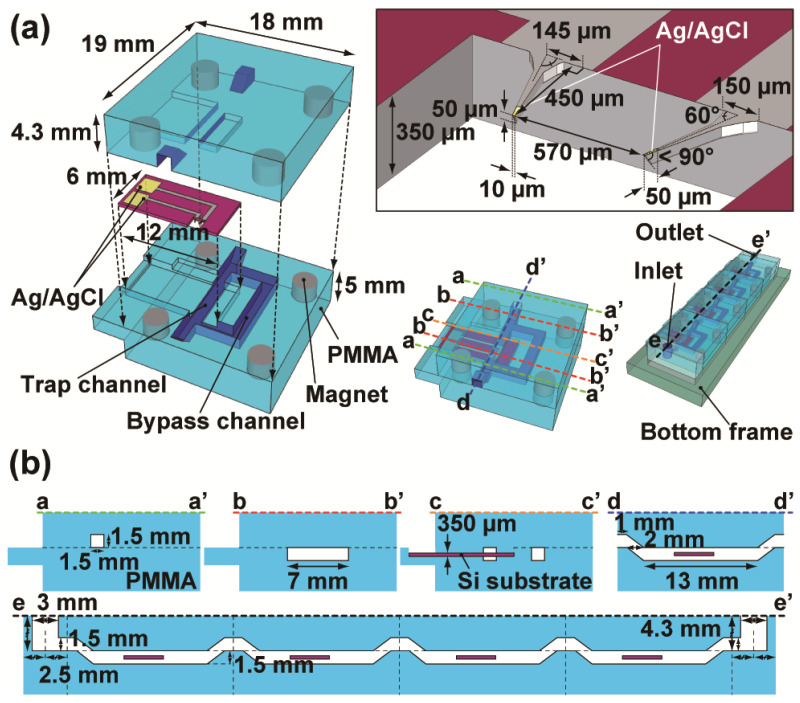
(**a**) Schematic images of the fluidic device with electrodes for an array of four oocytes. For surface insulating of the electrode chip, the chip was entirely covered with parylene-C, except for the contact pads and electrode tips (yellow areas). (**b**) Section images correspond to the colored dashed lines of a-a’, b-b’, c-c’, d-d’, and e-e’ in (**a**).

**Figure 2 sensors-23-02370-f002:**
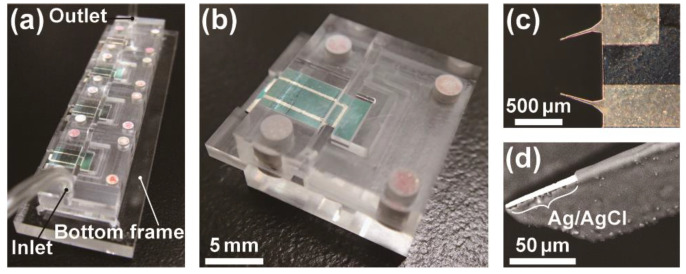
(**a**) Image of the device. (**b**) A single device unit. (**c**) Micrograph image (top view) of the electrodes. (**d**) Scanning electron microscopic image of an electrode tip.

**Figure 3 sensors-23-02370-f003:**
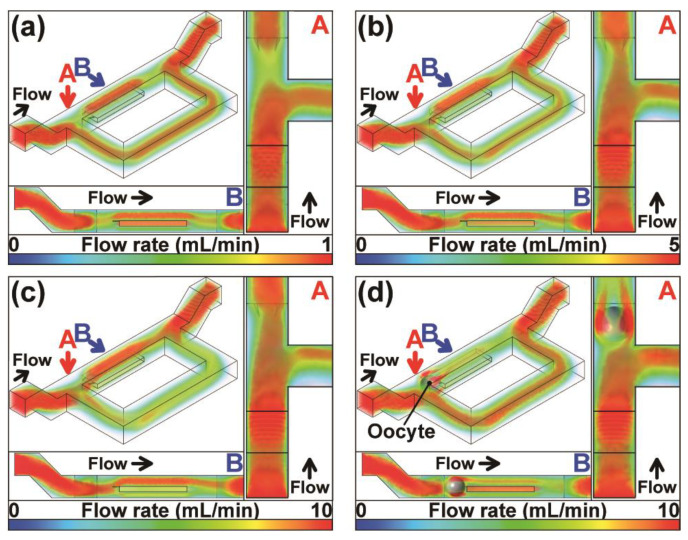
Stream renderings for fluid simulations at an entry flow rate of (**a**) 1 mL min^−1^, (**b**) 5 mL min^−1^, (**c**) 10 mL min^−1^ before oocyte trapping, and (**d**) 10 mL min^−1^ after oocyte trapping. Partial top and side views are shown in point of sights “A” and “B”, indicated by arrows in the upper left panels (overhead views).

**Figure 4 sensors-23-02370-f004:**
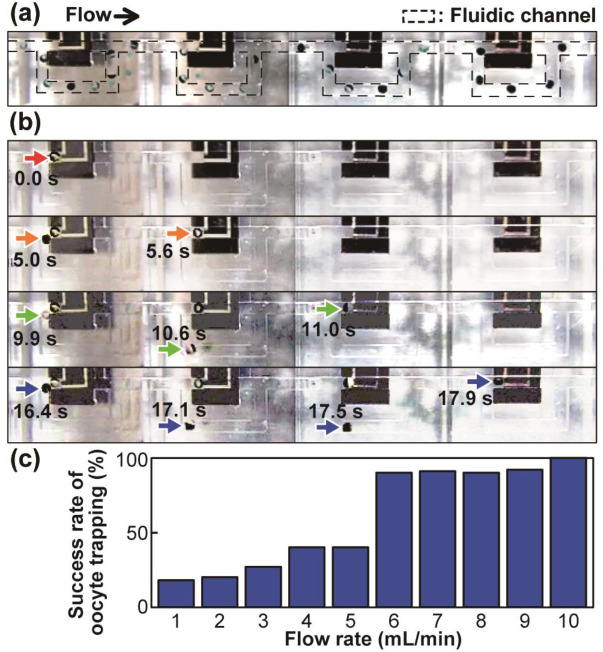
Superimposed images of oocyte motion; (**a**) an oocyte bypassed at a flow rate of 3 mL min^−1^, and (**b**) four different oocytes trapped at a flow rate of 20 mL min^−1^. The position of each oocyte over time is indicated by an arrow. (**c**) The success rate of oocyte trapping with respect to flow rate. Each data point is more than eight experiments.

**Figure 5 sensors-23-02370-f005:**
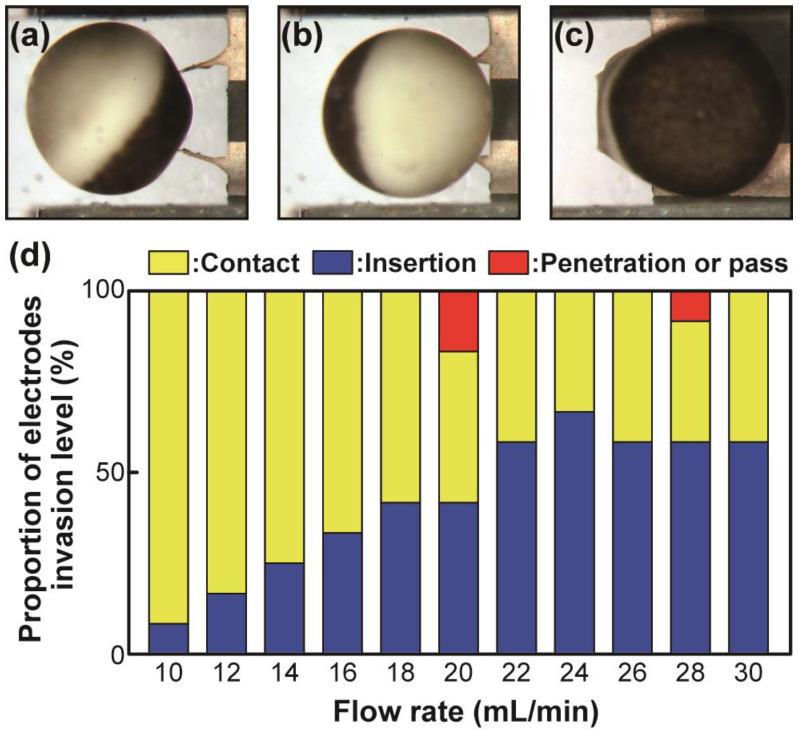
Photos of an oocyte that is (**a**) not inserted (failure), (**b**) inserted (success), and (**c**) penetrated by electrodes or passed through the trap (failure). Flow direction was left to right in (**a**), (**b**), and (**c**). (**d**) Correlation between the flow rate and electrode invasion level. Each data point represents 12 observations per flow rate.

**Figure 6 sensors-23-02370-f006:**
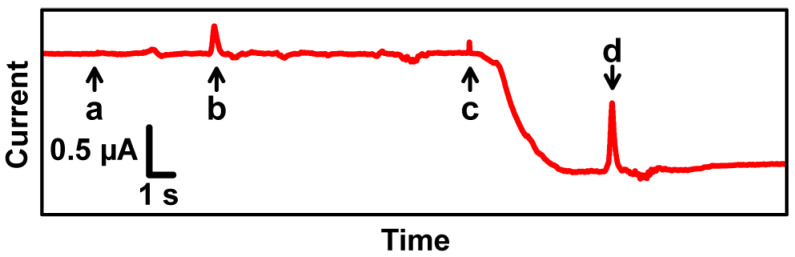
The current trace of an oocyte expressing BmOR3. At the time points indicated by arrows, each operation (addition or removal of 20 µL solution) was carried out sequentially. (**a**) Barth’s solution without bombykal was added. (**b**) The solution was removed from the fluidic device. (**c**) 100 µM bombykal solution was added. (**d**) The solution was removed again.

## Data Availability

Not applicable.

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
