# Peer review of "Xenopus laevis Oocyte Array Fluidic Device Integrated with Microelectrodes for A Compact Two-Electrode Voltage Clamping System"

_sensors, 2023, doi:10.3390/s23052370_

Round 1

Reviewer 1 Report

Please, see the file attached.

Author Response

The article by Misawa et al. is clearly written and well structured. It treats an actual problem of creating a compact and easy-to-operate chemical sensor device. The authors reported on using Xenopus oocytes expressing silkmoth olfactory receptor BmOR3 (that is sensitive to a silkmoth pheromone bombykal) in the developed fluidic chemical sensor. Current shifts corresponding to changes in Xenopus oocyte membrane potential caused by ligand irritation were measured by means of two-electrode voltage clamping (TEVC) method. Oocyte arrays and the appropriate insertion of electrodes were examined with respect to flow rate ranging from 1 to 30 mL/min. The videos in Supplementary files clearly demonstrate the oocyte passing or trapping mode depending on the flow rate.

One of the main goals of the study was compactization of sensor device by downsizing the TEVC system and simplifying the work with the sensor. The authors made a good job and developed a compact two-electrode voltage clamping system with a Si-based electrodes. In their previous paper (Ref.19), the authors described the odorant sensor based on a fluidic device employing oocytes that was shown to be highly sensitive and selective. The glass capillary electrodes were used in that device. As noted in the text, the insertion rate was lower when Si-based electrodes were used as compared to glass capillary electrodes and further electrode sharpening was required. Moreover, due to high concentration of bombykal solution used it was not able to test the lifetime of this sensor device by repeated chemical stimuli. So, my main question is whether it is possible to provide the additional characteristics of the sensor like sensitivity or selectivity in order to compare the newly-developed sensor with the previous version based on glass capillary electrodes?

Thank you for your comments and question. Unlike conventional glass capillary electrodes that are manufactured one by one, Si-based electrodes can be mass-produced simultaneously with the same characteristics. In addition, our research will make it possible to apply multiple oocytes to the sensor, and in the future, we believe that we will be able to detect a wider variety of chemical substances with higher sensitivity as we expand the variation of receptors to be expressed and a highly expressed system is constructed. Hence, we added the following sentences to the conclusion section.

(Line 276.)

Si-based electrodes are suitable for multi-channel measurement systems because they can be mass-produced simultaneously with uniform characteristics. In addition, since our fluidic devices have made it possible to apply multiple oocytes, we believe that in the future we will be able to detect multiple chemical substances with higher sensitivity by expanding the types of expression receptors and constructing highly expressing systems.

My minor comments are:

  1. It is noted in the Introduction, that chemical sensors based on biomaterials are suitable for short-term use. Could you please add some details regarding the estimated lifetime of the sensor based on Xenopus laevis oocyte?

Thank you for the helpful advice. In accordance with the reviewer's comment, the following sentences have been added.

(Line 258.)

Regarding the lifetime of the sensor using Xenopus oocytes, in the case of the glass capillary electrodes in our previous work [19], the activity was maintained for about half a day at room temperature. However, considering the current thickness of Si-based electrodes in this research, it is estimated that the lifetime of the sensor will be several hours at the longest, so we think that it is suitable for more temporary sensing at present.

  1. Figure 3. Is it possible to make some changes in the partial top and side views in Figure 3 (marked by red circles in Figure below) so as to make the flow directions in them coincide with flow directions in the main figures (as, for example, is depicted in Fig 3a*). I suppose, that such graphical representation would be easier-to-understand and comfortable for the readers (of course, if your initial representation is not an accepted standard for fluid simulations representation).

Thank you for pointing this out. As the reviewer said, we also thought the proposed arrangement would be easier for readers to understand. Therefore, we modified Figure 3 to the proposed arrangement.

Reviewer 2 Report

In the present study, the authors fabricated a two-electrode voltage clamping system for Xenopus laevis oocytes. The solution flow rate was tested by the fluid simulations and verified by oocytes experiments. Finally, the authors could successfully measure the response of oocytes to chemical stimuli. This interesting paper provides a novel insight into the electrode insertion of cell membrane and subsequent measurement that facilitates the high-throughput chemical sensing and screening field. The data are of high quality and the paper is clearly written and well reasoned. The references are also appropriately cited.

My minor comments are as follows:

1. Please add information about the manipulation of oocytes in the “ Materials and Methods” section, including the housing of Xenopus laevis, how to acquire/select gametes, and the loading of oocytes in the device.

2. Line 85, “Figure 1b”; line 168, “Figure 3a”.

3. In Figure 5d, legend “Penetration or pass”.

4. Line 254, “Xenopus laevis” should be italic.

Author Response

In the present study, the authors fabricated a two-electrode voltage clamping system for Xenopus laevis oocytes. The solution flow rate was tested by the fluid simulations and verified by oocytes experiments. Finally, the authors could successfully measure the response of oocytes to chemical stimuli. This interesting paper provides a novel insight into the electrode insertion of cell membrane and subsequent measurement that facilitates the high-throughput chemical sensing and screening field. The data are of high quality and the paper is clearly written and well reasoned. The references are also appropriately cited.

  1. Please add information about the manipulation of oocytes in the “Materials and Methods” section, including the housing of Xenopus laevis, how to acquire/select gametes, and the loading of oocytes in the device.

Thank you for your suggestion. We added the information about the content according to your advice.

(Line 129.)

The introduction of the oocytes into the fluidic device was performed by approaching an oocyte one-by-one with a micropipette manually to the inlet of tube connected to the device inlet side.

(Line 140.)

 Egg-bearing Xenopus laevis was raised in an incubator at 20 °C, and the oocytes were collected by laparotomy under ice-cold anesthesia.

  1. Line 85, “Figure 1b”; line 168, “Figure 3a”.

We are very sorry for our carelessness. We corrected the part you pointed out.

  1. In Figure 5d, legend “Penetration or pass”.

Thank you for pointing this out. We revised the word.

  1. Line 254, “Xenopus laevis” should be italic.

Thank you very much for your careful confirmation. We are sorry we didn't notice. We corrected it italics.
